# Flow Field Noise Analysis and Noise Reduction Research of Twin-Screw Air Compressor Based on Multi-Field Coupling Technology

Yayin He *, Xuyang He, Lijun Chen, Junli Wang, Yongqiang Zhao and Zhigui Ren

School of Mechanical Engineering, Shaanxi University of Technology, Hanzhong 723000, China; h06150616@163.com (X.H.); chenlijun4242024@163.com (L.C.)
* Correspondence: heyayin@snut.edu.cn

**Abstract:** To address the flow field noise problem in twin-screw air compressors, multi-physical-field coupling technology is employed to perform flow field noise calculations for the compressor. Based on the structural characteristics and noise generation principles of the twin-screw compressor, a noise reduction design method is proposed that employs a Helmholtz resonator and a three-chamber perforated muffler at the exhaust end. The muffler's structural optimization is performed using a genetic algorithm, and the effectiveness of the noise reduction method is validated through calculations. The results indicate that the Helmholtz resonator effectively mitigates airflow pulsation at the exhaust port, stabilizing the flow and reducing low-frequency noise at the exhaust end. Additionally, the designed three-chamber perforated muffler achieves noise reduction across a broad frequency range. With this noise reduction method, the exhaust port noise of the twin-screw compressor is reduced from 100–114 dB to 37–68 dB. These findings provide valuable insights for vibration and noise reduction in twin-screw air compressors.

**Keywords:** twin-screw air compressor; flow field noise; exhaust muffler; noise reduction studies





## 1. Introduction

Positive displacement compressors mainly include screw type, rotary type and reciprocating type. The twin-screw air compressor is a type of positive displacement compressor, which changes the volume of the internal chamber of the casing through the rotation of the negative rotor and positive rotor, and it completes with the process of suction, compression and exhaust. Because of its simple structure, stable performance and high efficiency, screw compressors have a wide range of applications in industrial production. As these compressors continue to improve and increase in performance, mechanical noise due to frictional wear and tear during operation has become a significant issue. Milojevit S et al. [1] addressed this by reducing friction and wear through optimized reciprocating compressor structures. In addition to mechanical noise, air compressors generate significant aerodynamic noise during both the suction and exhaust processes while operating. Therefore, researching and reducing compressor noise have become crucial goals in compressor technology. It is also a key factor for various compressor manufacturers aiming to enhance competitiveness.

Both the theory and application of noise control have matured domestically and internationally, and research on the noise issues of twin-screw air compressors has been steadily increasing. Sun Shizhong [2] conducted experimental tests on the exhaust pressure pulsation phenomenon of the twin-screw refrigeration compressor and investigated the influence of rotational speed and condensing temperature on the exhaust pressure pulsation. Wang Xiaoming et al. [3] conducted a numerical analysis of the exhaust characteristics of a balanced twin-screw compressor, examining the variations in exhaust airflow. Their findings offered important theoretical support for optimizing the exhaust port design.

Huang et al. [4] applied the theory of excitation wave management applied to the gas pulsation in the transient exhaust of a screw compressor, revealing that the nature of the gas pulsation is a combination of large-amplitude compression and expansion waves, accompanied by the phenomenon of induced fluid flow. A A G Abdulrahm [5] analyzed the noise emissions from jet engine intake to the compressor and nozzle exhaust and found that the main frequency of the exhaust noise generated by the jet discharge was linearly proportional to the axial speed of the compressor. Sun et al. [6] conducted tests on pressure fluctuation, noise characteristics and thermal performance for screw refrigeration compressors with two different modes of slide valve and inverter control, and the results showed that the inverter-controlled screw compressor had better overall performance. He et al. [7] studied the noise and vibration characteristics of the twin-screw air compressor based on the acousto-structure interaction method and found that, when the frequencies are different, the denser the acceleration distribution on the surface of the shell, the larger the sound pressure level. When the frequency is the same, with an increase in the incident pressure wave, the deformation of the shell also increases, and the maximum deformation and the maximum sound pressure level of the shell appear at about 2000 Hz. D. Guerrato et al. [8] measured the axially averaged flow and turbulent fluctuations in several cross-sections inside the rotor chamber of a screw compressor with high temporal and spatial resolution to validate a CFD model of the fluid flow inside a twin-screw compressor, which provides a basis for the optimal design of the compressor.

A Fujiwara et al. [9] used a comprehensive waveform analysis method to study the vibration noise of the oil-injected screw compressor, which provides theoretical support for studying the vibration noise of the compressor. J Willie et al. [10] used the finite element simulation method to optimize the design of manifold pulsation valves for the noise and vibration of the oil-less screw compressor, and the results of the study showed a better vibration and noise reduction effect. Zhang Weimin et al. [11] concluded, through experimental testing, that the compressor is the primary noise source in a screw refrigeration compressor unit. They proposed several noise reduction measures, including optimizing the rotor profile, adjusting the meshing gap between the negative and positive rotors, and improving rotor machining accuracy. Li Jinlu et al. [12] optimized the rotor profile of a fuel cell twin-screw air compressor, and the noise of the screw air compressor with the optimized rotor profile was significantly lower than that before optimization. Shen Jiubing et al. [13] optimized the flow path of the exhaust chamber of a semi-hermetic screw refrigeration compressor, which effectively reduced the compressor exhaust pressure pulsation and lowered the noise. Liu Huan et al. [14] chose and designed the acoustic enclosure to perform noise reduction analysis for the noise of the screw compressor unit.

Cheng Shuangling et al. [15] analyzed the noise source of a twin-screw compressor, conducted noise tests and reduced the compressor noise by installing check valves and acoustic enclosures. Jijunhe et al. [16] proposed a passive noise reduction method of adding damping material to the acoustic enclosure for the prominent low-frequency noise of the screw compressor, which improved the control of low-frequency noise via the acoustic enclosure. Niu Qunfeng et al. [17] improved the compressor air inlet structure and change the method of sound-absorbing materials to control the noise, to solve the problem of a certain model of the screw compressor noise exceeding the standard. He et al. [18] took the two-stage screw refrigeration compressor as an experimental object and tested and analyzed different working conditions of the compressor by using noise reduction methods such as acoustic wave interference and gas–solid resonance, and the results showed that it had a good noise reduction effect. Mujic E et al. [19] reduced the noise by improving the design of the shape of the exhaust port of the screw compressor. Shen J et al. [20] proposed two noise reduction methods for a semi-hermetic inverter screw refrigeration compressor, namely, end-face attenuation channel and discharge tube damping, and verified both noise reduction methods, which achieved a better noise reduction effect.

In summary, recent domestic research on the noise of twin-screw air compressors has primarily focused on optimizing screw rotor profiles, designing exhaust port structures and

improving system controls. In contrast, there has been relatively less research on reducing noise by incorporating mufflers. In this study, the noise characteristics of the suction and exhaust ports of a twin-screw air compressor are analyzed by calculating the flow field and noise within the compressor casing. The exhaust port noise is reduced using a unique design approach that incorporates a stabilizer Helmholtz resonator and a three-chamber perforated muffler. This study employs acoustic flow coupling to examine the steady flow within the Helmholtz resonator. Additionally, the structure of the three-chamber perforated muffler is optimized using a genetic algorithm. Noise reduction analysis is performed by installing the optimized muffler at the exhaust port of the air compressor, resulting in improved noise reduction performance.

## 2. Twin-Screw Compressor Flow Field Noise Calculation Method

### 2.1. Flow Field Analysis Methods

The gas flow within the internal passages of a twin-screw air compressor adheres to fundamental fluid dynamics equations, including the principles of mass conservation, momentum conservation and energy conservation, as detailed in [21].

(1) Mass conservation equations

$$\frac{\partial \rho}{\partial t} + \frac{\partial (\rho u)}{\partial x} + \frac{\partial (\rho v)}{\partial y} + \frac{\partial (\rho w)}{\partial z} = 0 \tag{1}$$

Let $div(a) = \partial a_x/\partial x + \partial a_y/\partial y + \partial a_z/\partial z$; the above equation can be expressed as:

$$\frac{\partial \rho}{\partial t} + div(\partial u) = 0 \tag{2}$$

Expressed in the form of divergence, then, $\nabla \cdot (a) = div(a) = \frac{\partial a_x}{\partial x} + \frac{\partial a_y}{\partial y} + \frac{\partial a_z}{\partial z}$, and then:

$$\frac{\partial \rho}{\partial t} + \nabla \cdot (\rho u) = 0 \tag{3}$$

where $\rho$ is the density, $t$ is the time and $u$, $v$, $w$ are the components of the velocity vector in the $x$, $y$ and $z$ directions, respectively.

(2) Momentum conservation equation

$$\begin{cases} \frac{\partial (\rho u)}{\partial t} + \nabla \cdot (\rho u V) = \frac{\partial \tau_{xx}}{\partial x} + \frac{\partial \tau_{yx}}{\partial y} + \frac{\partial \tau_{zx}}{\partial z} - \frac{\partial P}{\partial x} + F_x \\ \frac{\partial (\rho v)}{\partial t} + \nabla \cdot (\rho v V) = \frac{\partial \tau_{yy}}{\partial y} + \frac{\partial \tau_{xy}}{\partial x} + \frac{\partial \tau_{zy}}{\partial z} - \frac{\partial P}{\partial y} + F_y \\ \frac{\partial (\rho w)}{\partial t} + \nabla \cdot (\rho w V) = \frac{\partial \tau_{zz}}{\partial z} + \frac{\partial \tau_{xz}}{\partial x} + \frac{\partial \tau_{yz}}{\partial y} - \frac{\partial P}{\partial z} + F_z \end{cases} \tag{4}$$

where the vector operator is $\nabla \equiv \frac{\partial}{\partial x}\vec{i} + \frac{\partial}{\partial y}\vec{j} + \frac{\partial}{\partial z}\vec{k}$; P is the pressure on the fluid microelement; $\tau_{xx}$, $\tau_{xy}$, $\tau_{xz}$, etc., are the components of the viscous fluid molecules generating viscous forces on the surface of the microelement body; $Fx$, $Fy$ and $Fz$ are the forces acting on the microelement body. If there is only gravity and in the opposite direction to $+z$, there is $F_x = F_y = 0$, $F_z = -\rho g$.

(3) The energy conservation equation

$$\frac{\partial (\rho T)}{\partial t} + div(\rho u T) = div\left(\frac{k}{C_p} grad T\right) + S_T \tag{5}$$

Expanding Equation (5) yields Equation (6)

$$\frac{\partial (\rho T)}{\partial t} + \frac{\partial (\rho u T)}{\partial x} + \frac{\partial (\rho v T)}{\partial y} + \frac{\partial (\rho w T)}{\partial z} = \frac{\partial}{\partial x}\left(\frac{k}{C_p}\frac{\partial T}{\partial x}\right) + \frac{\partial}{\partial y}\left(\frac{k}{C_p}\frac{\partial T}{\partial y}\right) + \frac{\partial}{\partial z}\left(\frac{k}{C_p}\frac{\partial T}{\partial z}\right) + S_T \tag{6}$$

where $C_P$ is the specific heat capacity, $T$ is the temperature, $k$ is the fluid heat transfer coefficient and $S_T$ is the viscous dissipation term.

### 2.2. Sound Field Analysis Methods

(1)    Equation for acoustic waves in fluid media:

Acoustic waves propagate as plane waves in regularly shaped pipes, while they propagate as three-dimensional waves in irregularly shaped cavities. The equations governing plane waves and three-dimensional waves in a moving medium are presented in Equations (6) and (7), respectively [22].

$$\frac{\partial^2 p}{\partial x^2} - \frac{1}{c}\frac{\mathrm{D}^2 p}{\mathrm{D}t^2} = 0 \tag{7}$$

$$\nabla^2 p - \frac{1}{c^2}\frac{\mathrm{D}^2 p}{\mathrm{D}t^2} = 0 \tag{8}$$

(2)    Helmholtz equation

In analyzing the noise generated by the twin-screw air compressor and the noise reduction effect of the muffler, the model is divided into several units to determine the noise distribution and obtain the noise spectrogram. The acoustic equations in the frequency domain are then solved as follows [22]:

$$\nabla \cdot (-\frac{1}{\rho_c}\nabla p) - \frac{\omega^2}{\rho_c c^2} = 0 \tag{9}$$

where $p$ is the pressure in units of $\mathrm{N \cdot m^{-2}}$, $\rho_c$ is the density in units of $\mathrm{kg \cdot m^{-3}}$, $\omega$ is the angular frequency in units of $\mathrm{rad \cdot s^{-1}}$ and $c$ is the speed of sound in units of $\mathrm{m \cdot s^{-1}}$.

## 3. Model Building and Solution Setup

### 3.1. Modelling of the Flow Field

In this study, according to the fluid analysis steps, the 3D model of the air compressor flow field is imported into the finite element simulation software, and the meshing, boundary condition setting and post-processing of the results are carried out. The research object of this study is a twin-screw air compressor with screw rotor model LGY03 in a factory, and its shell model is shown in Figure 1a. The establishment of the flow field model of the compressor is the basis for flow field simulation and analysis: firstly, the shell model of the compressor is established, and then the interior of the shell model is filled to obtain the filled model, and the screw rotor in the filled model is subtracted by Boolean operation to obtain the flow field model of the compressor. Figure 1b shows the assembly of the air compressor rotor and flow field model, and Figure 1c shows the air compressor flow field model.

### 3.2. Mesh Division of Air Compressor Flow Field

COMSOL Multiphysics is a powerful software tool for simulation calculations, capable of performing dual-physics and multiphysics coupling calculations. It includes various modules, such as fluid, acoustics and structural modules, allowing for multi-field coupling calculations both within and between different physical fields. This makes it particularly convenient for solving complex multiphysics coupling problems. The Acoustics Module includes various features, such as pressure acoustics, acoustic–structure interactions, aeroacoustics, thermoviscous acoustics, ultrasonics, and geometrical acoustics, all of which can be calculated using different acoustic interfaces. According to the research object of this study, the COMSOL 5.6 version of the pressure acoustics module and the acoustic field and flow field coupling module are used.

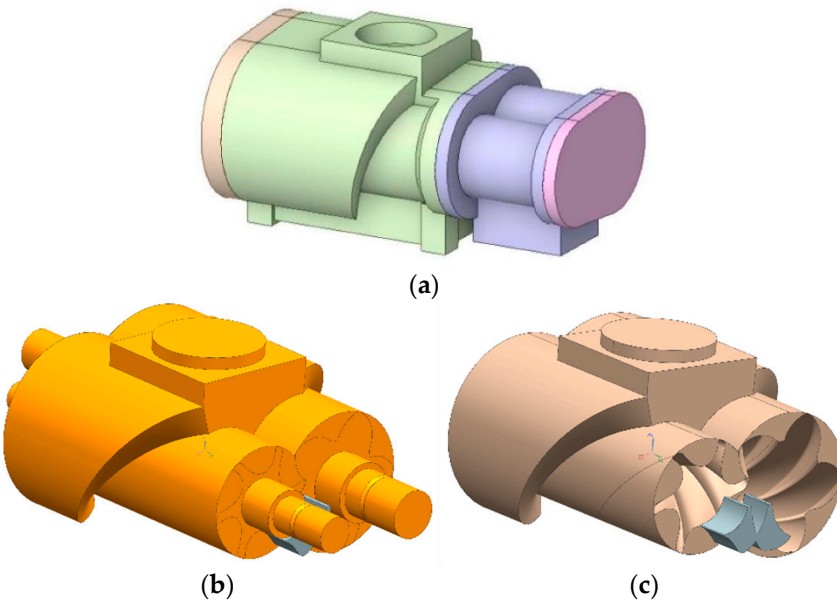

**Figure 1.** Sound structure coupling sound field model. (**a**) Air compressor shell model, (**b**) the assembly of the air compressor rotor and flow field model, (**c**) air compressor flow field model.

The flow field model of the twin-screw compressor is meshed by first dividing the model into sections based on the working principle. These sections include the air inlet, air outlet, negative rotor wall, positive rotor wall and other-wall surfaces, each of which is then appropriately named. Due to the existence of complex spiral surfaces in the flow field model, tetrahedral cells are used for meshing, using the patch conformal method, with a mesh size of 2.5 mm. Due to the complex structure and rotating state of the flow field region in contact with the walls of the negative and positive rotors, the surface mesh in this area is refined to a mesh size of 1.5 mm. This refinement results in 522,841 mesh nodes and 2,686,392 mesh cells. The quality of the generated mesh is then checked to ensure it meets the simulation requirements. The meshed flow field model of the air compressor is shown in Figure 2.

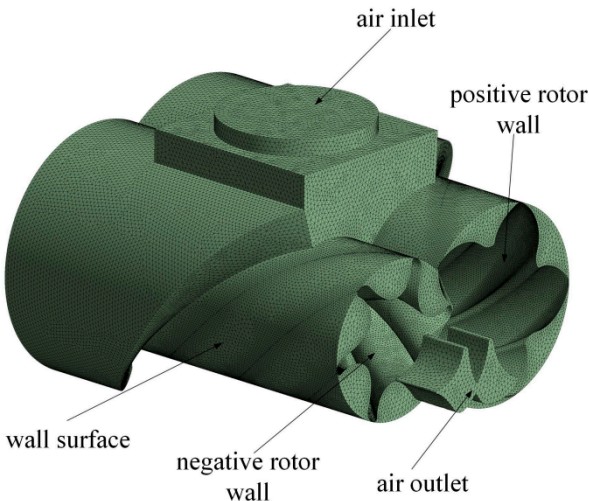

**Figure 2.** Air compressor flow field model meshing.

### 3.3. Setting of Boundary Conditions

Since the twin-screw air compressor compresses air through the meshing of the negative and positive rotors, it is essential to define the boundaries for the suction port pressure, exhaust port pressure, contact walls of the negative and positive rotors, and rotational

speed, among other parameters. The pressure at the compressor's suction port is 0.1 Mpa, while the pressure at the exhaust port is 0.8 Mpa. The suction temperature is 20 °C, and the exhaust temperature is 80 °C. The ratio of the number of teeth on the compressor's negative and positive rotors is 6:5, and the ratio of the rotational speeds of the negative and positive rotors is 1:1.2. The rotation speed of the compressor is 5000 r/min, and the rotation speed of the wall surface in contact with the positive rotor is 5000 r/min; the rotation speed of the rotating wall surface in contact with the negative rotor is 4167 r/min, and the wall surface outside the flow field is static. A pressure–velocity coupling scheme was adopted using the k-epsilon standard turbulence model.

## 4. Twin-Screw Compressor Flow Field and Noise Calculation

### 4.1. Flow Field Pressure Calculation

The finite element simulation software is used to calculate the established twin-screw air compressor model, and a pressure contour diagram of the air compressor flow field is obtained, as shown in Figure 3. The finite element model of the twin-screw air compressor is calculated using the multiphysics coupling method, resulting in the pressure contour of the air compressor flow field. It is observed that the pressure value in the inlet basin is 0.12 Mpa, indicating that the pressure of the internal flow field is lowest at the air inlet. In the base element volumetric basin, as the negative and positive rotors engage to compress the air, the pressure in the flow field gradually increases. The pressure value is higher closer to the exhaust port, indicating that air is drawn in from the inlet, compressed by the rotating negative and positive rotors, and then discharged at the exhaust port. The pressure increases progressively and reaches a maximum value of 0.8 MPa at the exhaust port.

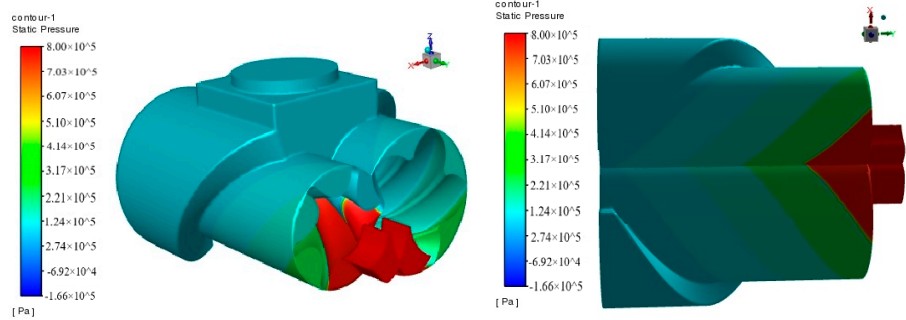

**Figure 3.** Pressure field cloud.

### 4.2. Flow Field Noise Analysis

The obtained flow field pressure and air compressor model are imported into the pressure acoustics module of the multiphysics field coupling software, and the flow field pressure is used as the input boundary condition of the acoustic field simulation to solve the noise value of the air compressor in the operating frequency band 20–5000 Hz. The maximum grid size for acoustic calculation cannot be larger than one-fifth of the wavelength (340 [m/s]/5000 [Hz]/5). In accordance with acoustic mesh requirements, the mesh obtained from the finite element simulation is mapped onto the acoustic mesh. The mesh quality is then analyzed using the multiphysics field software to ensure it meets the computational requirements. In this study, the air compressor flow field model is defined with the suction port on the upper end face as the inlet for acoustic wave transmission and the exhaust port on the right end face as the outlet. The inlet pressure is set at 0.1 MPa and the exhaust pressure at 0.8 MPa. The study material is defined as air with a density of 1.225 kg/m$^3$, a speed of sound of 340 m/s, a suction temperature of 20 °C and an exhaust temperature of 80 °C. The model is then used to calculate the noise levels at the air compressor's suction and exhaust ports under various operating conditions. Figure 4 displays the sound pressure distribution of the air compressor flow field domain at a

frequency of 2680 Hz, while Figure 5 shows the sound pressure level distribution at a frequency of 3500 Hz.

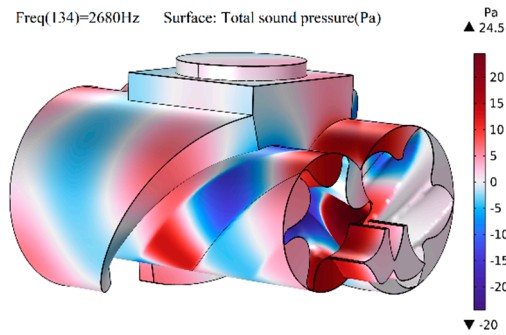

**Figure 4.** Sound pressure diagram of the flow field at 2680 Hz.

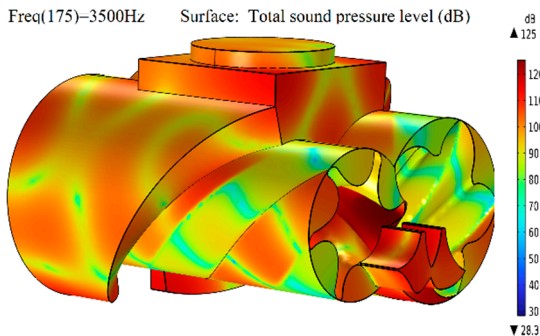

**Figure 5.** Sound pressure level plot at 3500 Hz in the flow field domain.

In Figures 4 and 5, at a frequency of 2680 Hz, the maximum sound pressure in the flow field reaches 24.5 Pa, concentrated predominantly at the exhaust port and in proximity to the engagement zone between the negative and positive rotors. The minimum sound pressure occurs mainly at the rotating walls of the rotors, with a minimum value of −20 Pa. Additionally, there is a distribution of positive and negative sound pressures in the inlet area, ranging from −10 Pa to 10 Pa.

At a frequency of 3500 Hz, the maximum noise is concentrated at the exhaust port. Additionally, the cloud map shows darker areas near the engagement zone of the negative and positive rotors close to the exhaust port, indicating higher noise levels in this region. The noise can reach up to 125 dB. This phenomenon is primarily due to the rotation of the negative and positive rotors, which compresses the air. The high-pressure gases are intermittently discharged through the smaller exhaust port, resulting in airflow pulsations that generate increased noise. In the flow field domain, turbulence generated by the compression and rotation of airflow at the meshing of the negative and positive rotors contributes to increased noise levels. At a frequency of 2680 Hz, the maximum sound pressure in the flow field reaches 24.5 Pa, primarily concentrated at the exhaust port and near the engagement zone of the yin and yang rotors. The minimum sound pressure is observed on the rotating walls of the negative and positive rotors, with a minimum value of −20 Pa. Additionally, sound pressures in the inlet basin vary between −10 Pa and 10 Pa, exhibiting both positive and negative distributions.

*4.3. Numerical Calculation of Noise at the Suction and Exhaust Ports*

The flow field of the air compressor was analyzed, focusing on the suction and exhaust ports within the compressor's flow domain. The noise spectra of the suction and exhaust ports of the compressor at an exhaust pressure of 0.8 MPa and a frequency range of 20–5000 Hz are shown in Figures 6 and 7, respectively.

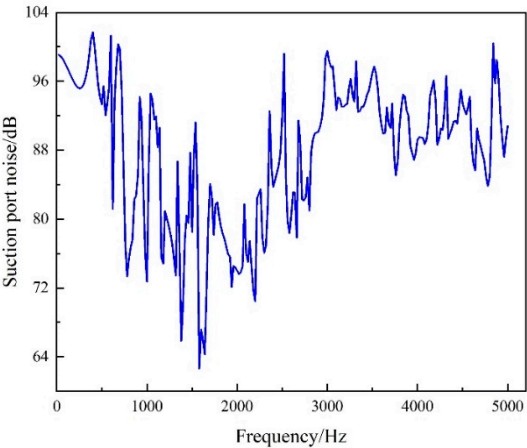

**Figure 6.** Noise spectrum at the suction port of the air compressor.

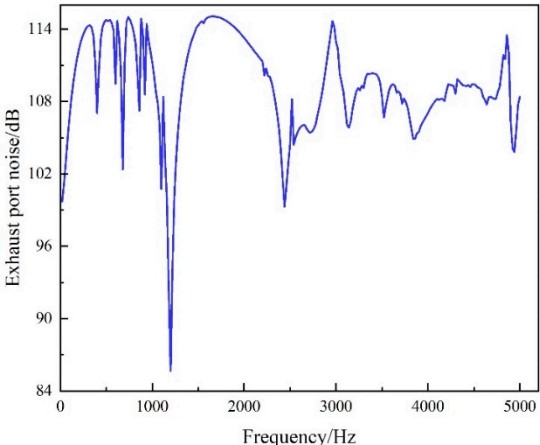

**Figure 7.** Noise spectrum at the exhaust port of the air compressor.

Figure 6 shows that the noise levels at the suction port of the air compressor range from 62 to 102 dB. The highest noise level, 101.7 dB, occurs at a frequency of 400 Hz, while the lowest, 62.6 dB, is observed at 1580 Hz. Noise is predominantly concentrated within the 70–100 dB range. From the start of the compressor's operation until the power stabilizes, the noise at the suction port initially decreases, then increases, and eventually stabilizes, with most noise levels falling between 84 and 100 dB. Figure 7 shows that the noise at the exhaust port is significantly higher than at the suction port, with noise levels primarily concentrated between 100 and 114 dB. The lowest noise level at the exhaust port, 85.7 dB, occurs at a frequency of 1200 Hz, while the highest, 115 dB, is observed at 1660 Hz. At the same time, there is a large noise generation at 320 Hz, 500 Hz, 740 Hz and 2960 Hz.

By reducing the noise of the suction and exhaust ports of the twin-screw compressor, it can be seen that the noise generated at the exhaust port of the compressor is significantly greater than the noise of the suction port. To mitigate noise pollution and reduce the associated hazards of the air compressor, it is essential to implement noise reduction measures. Therefore, the fifth part of this study focuses on achieving noise reduction by designing a muffler for the exhaust port.

### 4.4. Experimental Research and Analysis

To verify the accuracy of the noise and noise reduction simulation for the flow field of the twin-screw air compressor, a twin-screw air compressor with identical technical parameters and working conditions to the model constructed in this study was used as the test subject. The noise levels at the suction port were measured across a frequency range of 20–2000 Hz, and the results were compared with the simulation outcomes. The whole test

system consists of an air filter, air compressor, belt, electric motor and other supporting equipment. A test system site diagram is shown in Figure 8.

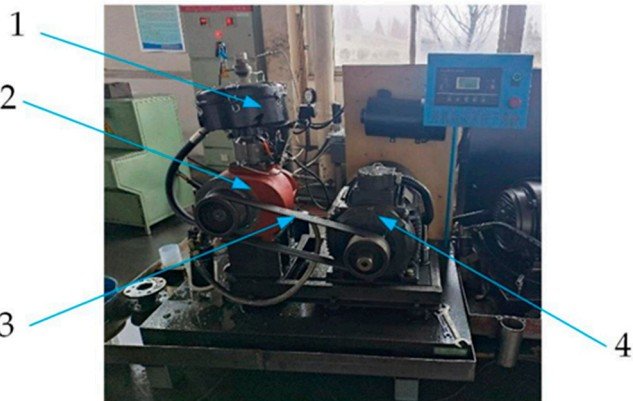

**Figure 8.** Test system site diagram 1. Air filter 2. Air compressor 3. Belt 4. Electric motor.

(1)    Principle of operation of the test system

Before conducting the experiment, a no-load test was performed, followed by the experimental test after confirming the no-load test results were satisfactory. The motor was started, causing the compressor to begin rotating, compressing the air. The compressed oil and gas mixture was expelled from the exhaust port and entered the oil and gas separator. After separation in the oil and gas separator, high-pressure air and oil were obtained. The high-pressure air passed through the minimum pressure valve, with part of it recirculated back into the air compressor, while the rest was cooled by the cooler before being discharged and recycled back into the main body of the air compressor. To collect the necessary data for the test, a noise meter was used to measure noise levels at various frequencies, positioned 0.2 meters above the suction port of the screw compressor (point A). Figure 9 illustrates the layout of the measurement point at the compressor's suction port.

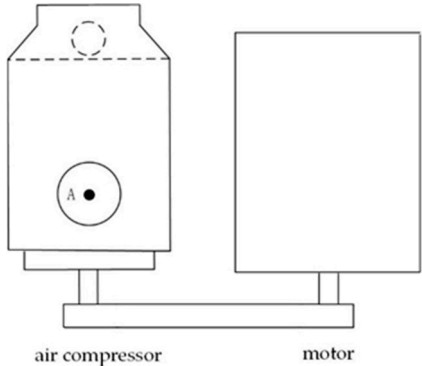

**Figure 9.** Layout of air compressor suction port noise measurement point.

(2)    Comparative analysis of test data and simulation results

To validate the accuracy of the simulation, the noise at the suction port of the twin-screw compressor was measured at an exhaust pressure of 0.8 Mpa, with a frequency range of 20–2000 Hz. The measurement results at different frequencies are presented in Table 1 below. Figure 10 compares the test and simulation noise levels, showing a maximum error of 6.8% between the simulation and the test points, which indicates that the simulation is reasonable.

**Table 1.** Suction port noise test values.

| Frequence/Hz | Noise/dB | Frequence/Hz | Noise/dB |
|---|---|---|---|
| 69 | 102.5 | 307 | 89.3 |
| 87 | 99.7 | 468 | 89.8 |
| 105 | 95.8 | 565 | 91.3 |
| 132 | 94.0 | 712 | 89.0 |
| 160 | 92.6 | 859 | 84.3 |
| 202 | 93.4 | 1079 | 89.7 |
| 244 | 91.5 | 89.7 | 77.3 |

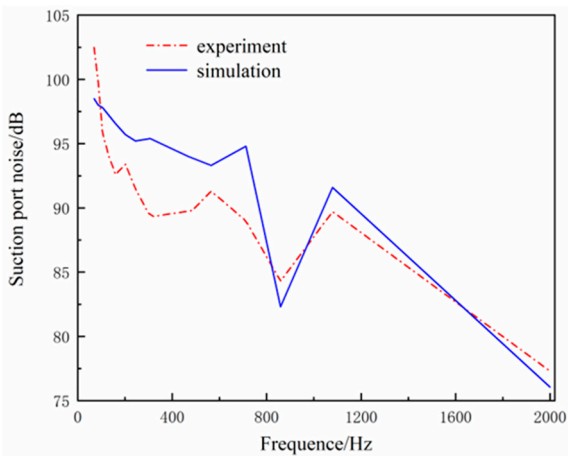

**Figure 10.** Comparison of test and simulation results.

## 5. Exhaust Muffler Design and Air Compressor Noise Reduction Research

As shown in Figure 7, the noise at the exhaust port of the air compressor is highest in the mid- to low-frequency bands, with several extreme values. Although the noise levels in the high-frequency band are lower than those in the low-frequency band, they still remain significant. The exhaust process of the air compressor is cyclic, and the high exhaust airflow velocity can cause pressure pulsations, leading to significant pressure pulsation noise. Additionally, this large exhaust noise can induce vibrations in the exhaust pipeline, generating pipe vibration noise. To mitigate these issues, a Helmholtz resonator is designed to be installed at the compressor's exhaust port, acting as a flow stabilizer to reduce exhaust gas velocity and lower the low-frequency exhaust noise. Furthermore, an impedance composite muffler is designed to reduce noise across the entire frequency range.

### 5.1. Design of Helmholtz Resonator

The Helmholtz resonator is a classic low-frequency muffler, comprising a thin tube attached to the main pipe and a closed cavity connected to the thin tube. The diameter of the thin tube is smaller than that of the main pipe, while the diameter of the closed cavity is much larger than the main pipe's diameter, both directly and when bypassing the thin tube. When the sound transmission frequency matches the resonator's resonance frequency, the resonator's side cavity will short-circuit the frequency band of the sound wave. This effectively blocks the sound wave from continuing forward in the transmission process, acting as a filter within the pipeline and achieving better noise reduction. Figure 11 shows a two-dimensional diagram of the designed Helmholtz resonator, which is connected to the main pipe through the bypass thin tube.

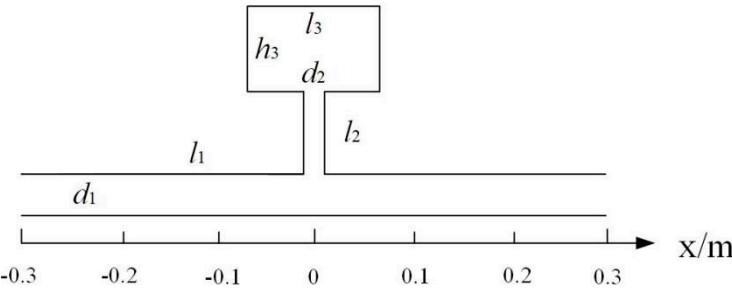

**Figure 11.** Helmholtz resonator in two dimensions.

In order to achieve a better noise reduction effect, through the calculation of the resonator's anechoic volume [23], and combined with the assembly space dimensions of the air compressor, the design parameters of the Helmholtz resonator are determined as follows: the main pipe diameter $d_1$ is 20 mm and the length $l_1$ is 600 mm; the thin pipe diameter $d_2$ is 16 mm and the length $l_2$ is 40 mm; the length $l_3$ of the rectangular cavity is 100 mm, and the height $h_3$ is 80 mm.

*5.2. Calculation of Muffler Transmission Loss*

Before calculating the coupled flow and acoustic fields for the Helmholtz resonator, the meshes for both fields are resolved. First, the flow field within the muffler is analyzed using the turbulence module. The internal fluid domain is defined as air, with the inlet and outlet of the muffler specified. The inlet velocity is set as UinU_{in}Uin. A free quadrilateral mesh is used, with mesh properties defined as hydrodynamic, a maximum mesh size of 5 mm, and a minimum mesh size of 1 mm. The Helmholtz resonator comprises 678,751 mesh elements.

After completing the fluid flow solution, a computational grid for the acoustic field is added. A PML (Perfectly Matched Layer) is set up at the inlet and outlet of the Helmholtz resonator. A plane wave with a pressure amplitude of 1 Pa is input at the inlet. The grid type is a free tetrahedral grid, with a maximum grid size equal to one-eighth of the main pipeline's diameter and a minimum grid size equal to one-fifteenth of the main pipeline's diameter. A multiphysics field coupling study is then conducted, mapping the flow field grid solution onto the acoustic field grid. The mesh quality is analyzed using multiphysics field software to ensure it meets simulation requirements. The mesh division is illustrated in Figure 12.

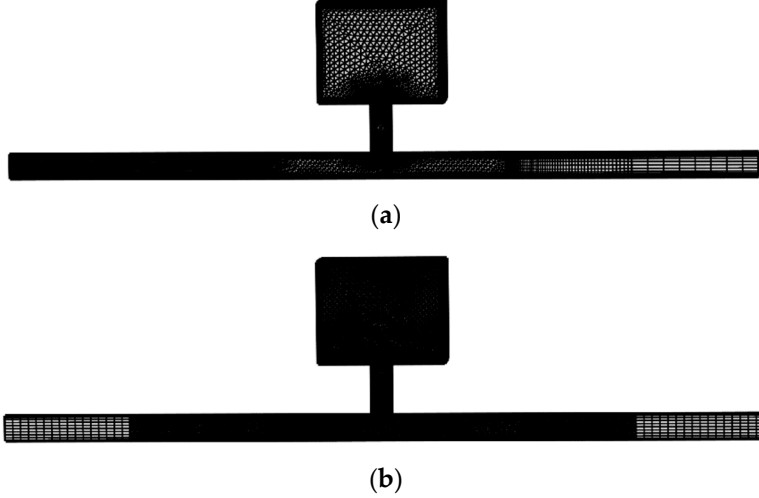

**Figure 12.** Helmholtz resonator meshing. (**a**) Helmholtz resonator flow field mesh, (**b**) Helmholtz resonator sound field mesh.

To solve the attenuation effect of the Helmholtz resonator on the airflow through it, the velocity at the inlet of the resonator is defined as Uin = C0Ma, and the velocity of the airflow at the exhaust port is calculated to be 20–50 m/s. Ma = 0.06, Ma = 0.09 and Ma = 0.12 were chosen to test the attenuation effect of the resonator on the airflow. Figure 13 shows a turbulent viscosity cloud diagram of the Helmholtz resonator at Mach number 0.12, where the exhaust flow enters from the left side of the main pipe, and at this time, the color of the inside of the pipe on the left side is dark red, which indicates that the turbulent viscosity value of the gas flow just entering the main pipe is larger, the turbulence phenomenon is more obvious, and the turbulence noise generated is also larger. As the gas flow passes through the resonator, most of it enters the resonance cavity through the thin tube at the neck of the resonator, reducing the turbulence of the gas. This is reflected by the color changing to light green and blue. A smaller portion of the gas exits through the right exhaust pipe, where turbulence is further reduced, and the turbulent viscosity values are lower compared to those of the gas entering the main pipe.

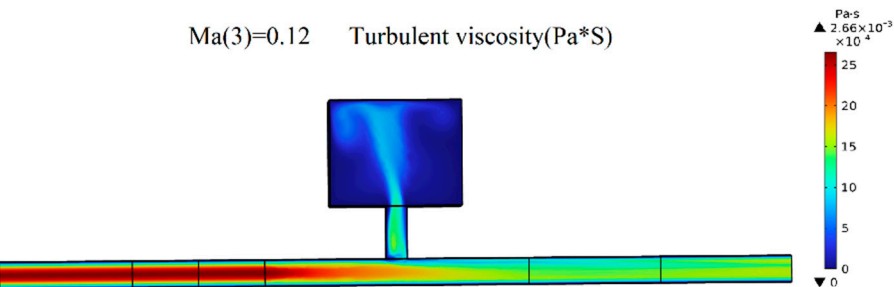

**Figure 13.** Helmholtz resonator meshing.

The change in turbulent viscosity along the main pipe of the Helmholtz resonator is illustrated in Figure 14. The figure shows that the turbulent viscosity of the airflow entering the main pipe remains largely unchanged. However, as the airflow passes through the resonator, the turbulent viscosity steadily decreases, indicating a significant reduction in turbulence. There is a slight increase in turbulent viscosity as the airflow approaches the exhaust port. Overall, after the resonator, the turbulent viscosity in the pipeline is greatly relieved, indicating that the resonator has a better stabilizing effect on the gas.

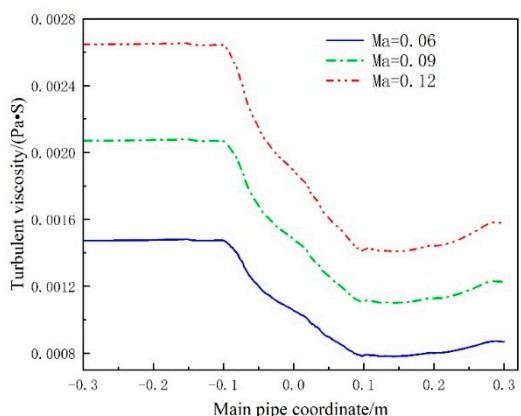

**Figure 14.** Turbulent viscosity in the main pipe of the Helmholtz resonator.

Since the Helmholtz resonator is a low-frequency muffler with effective noise reduction in the low-frequency band, its frequency range is determined to be 20–500 Hz. The transfer loss of the Helmholtz resonator at different Mach numbers within this frequency range is shown in Figure 15. The figure indicates that the transfer loss is influenced by the Mach number. Specifically, in the 20–260 Hz range, the Mach number significantly affects the transfer loss: as the Mach number increases, the transfer loss decreases, and conversely, as

the Mach number decreases, the transfer loss increases. When the frequency is between 340 and 500 Hz, the Mach number has no significant effect on the transmission loss value. In this range, as the frequency increases, the transfer loss decreases. Thus, after the airflow passes through the Helmholtz resonator, it effectively reduces turbulence and the noise caused by pipeline vibration in the low-frequency band, contributing to better flow stabilization. However, in the mid- to high-frequency bands, the noise reduction effect is less pronounced. The twin-screw compressor generates significant noise across both the low-frequency and mid- to high-frequency bands. To effectively reduce noise across the entire frequency spectrum, a muffler is also required for noise reduction.

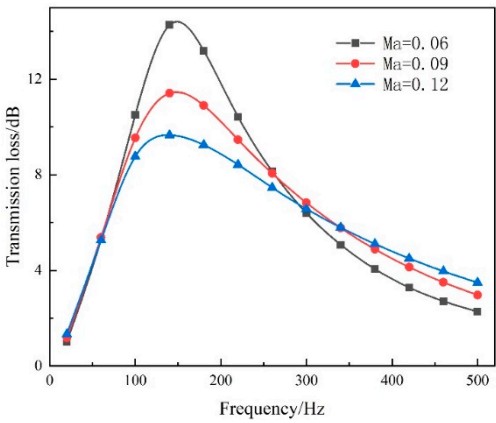

**Figure 15.** Transfer loss diagram for Helmholtz resonators.

### 5.3. Design of Three-Cavity Interpolated Perforated Muffler

To achieve better noise reduction in both the low-frequency and mid- to high-frequency bands, a three-cavity perforated muffler was designed, as shown in Figure 16. The muffler inlet and exhaust pipes are designed as an insertion type, which can reduce the cut-off frequency of the muffler and enhance the noise reduction effect, while the perforated form is designed on the inlet and exhaust pipes inserted into the muffler, which can improve the diffusion of the sound wave into the muffler. The muffler is designed with three chambers, each divided by an impedance plate in the middle to better attenuate sound waves during propagation. A connecting pipe is positioned centrally to minimize impact on the air compressor's performance. The perforations on the connecting pipe are designed similarly to those on the inlet and exhaust pipes, serving to diffuse the sound waves effectively.

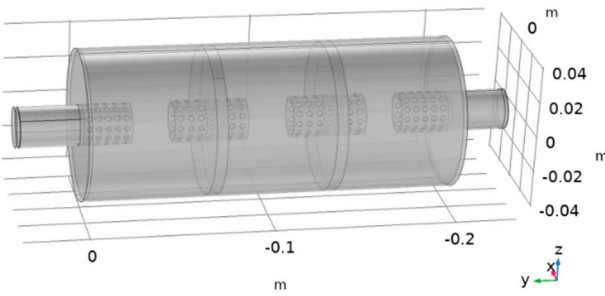

**Figure 16.** Three-cavity interpolated perforated muffler.

The muffler must fit within the internal space and assembly constraints of the air compressor unit. The design dimensions are as follows: the length of the inlet and exhaust pipes is 30 mm; the length of the perforated pipe inserted into the internal muffler is also 30 mm with a diameter of 20 mm; the impedance plate has a diameter of 80 mm and a thickness of 5 mm; the diameter of the connecting perforated pipe is 20 mm with a length of 40 mm; and the muffler cavity has a length of 200 mm.

A transmission loss diagram of the three-cavity perforated muffler is shown in Figure 17. The diagram indicates that this muffler effectively reduces the cut-off frequencies between 1220–3920 Hz and 3920–5000 Hz, significantly improving noise reduction performance. While the muffler provides high transmission loss in the mid- and high-frequency bands, it offers limited noise reduction in the low-frequency band, resulting in poorer and less stable noise reduction performance in that range.

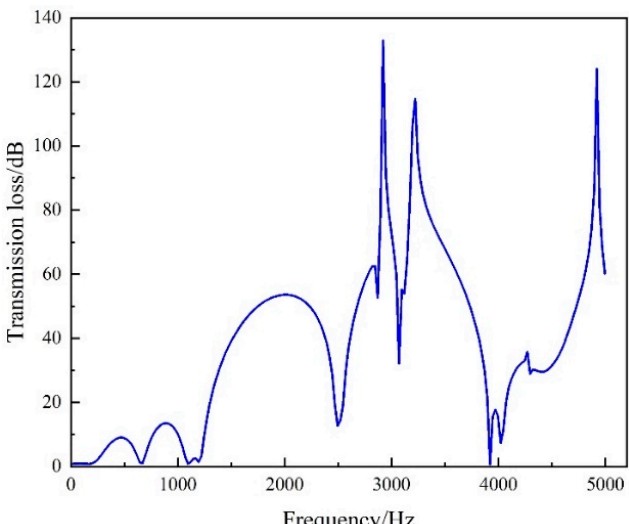

**Figure 17.** Three-cavity interpolated perforated muffler transmission loss.

### 5.4. Optimized Design of Three-Cavity Interpolated Perforated Muffler Structure

To enhance the noise reduction performance of the muffler, the structural dimensions of the three-cavity perforated muffler were optimized using a genetic algorithm.

(1) Optimization Objectives

A two-dimensional diagram of a three-chambered interpolated perforated muffler is shown in Figure 18, wherein $d_1$ is the diameter of the muffler suction pipe, which is aligned with the diameter of the exhaust pipe, $d_2$ is the diameter of the muffler cavity, $l_1$ is the length of the muffler suction insertion pipe, $l_2$ is the length of the muffler exhaust pipe of the first cavity, $l_3$ is the length of the suction pipe of the second cavity, $l_4$ is the length of the second cavity exhaust pipe, $l_5$ is the length of the suction pipe of the third cavity, $l_6$ is the length of the muffler exhaust insertion pipe, $l_7$ is the length of the first cavity, $l_8$ is the length of the second cavity, and $l_9$ is the total length of the muffler cavity.

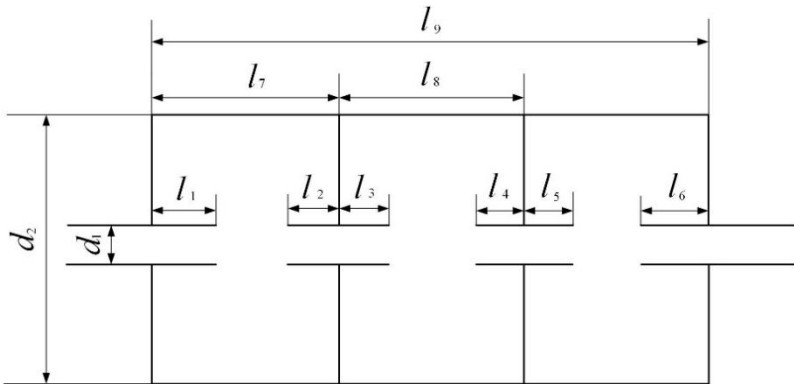

**Figure 18.** Two-dimensional drawing of three-cavity inserted perforated muffler.

According to muffler transfer matrix theory [24], mufflers are composed of basic acoustic units, and the total transfer matrix of the muffler is the product of these individual

units. Therefore, the muffler can be divided into three cavities, each of which can be decomposed into the insertion tube sudden expansion unit, the straight tube unit and the insertion tube sudden contraction unit, as illustrated in Figure 19a–c. The transfer matrices for each of these units are as follows:

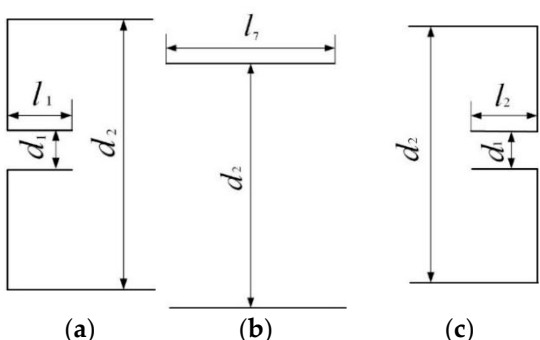

**Figure 19.** Basic anechoic unit: (**a**) the insertion tube sudden expansion unit, (**b**) the straight tube unit, (**c**) the insertion tube sudden contraction unit.

(1)  The insertion tube sudden expansion unit of the first cavity:

$$T_1 = \begin{bmatrix} 1 + \alpha_1 & -\frac{2M_1}{m_1\sigma}\left(1 - \frac{1}{\sigma}\right) \\ jm_2\beta_1\tan(kl_1) & 1 - \eta_1 \end{bmatrix} \tag{10}$$

where

$$\alpha_1 = j\frac{2m_2M_1}{m_1\sigma}\tan(kl_1), \beta_1 = \left[j\frac{m_2\left(1-\frac{2}{\sigma}\right)M_1}{m_1}\tan(kl_1) - 1\right],$$
$$\eta_1 = j\frac{m_2}{m_1}M_1\left(1 - \frac{2}{\sigma}\right)\tan(kl_1)$$

(2)  The straight tube unit of the first cavity

$$T_2 = \begin{bmatrix} \cos\frac{kl_7}{1-M^2} & \frac{j}{m}\sin\frac{kl_7}{1-M^2} \\ jm\sin\frac{kl_7}{1-M^2} & \cos\frac{kl_7}{1-M^2} \end{bmatrix} \tag{11}$$

(3)  The insertion tube sudden contraction unit of the first cavity:

$$T_3 = \begin{bmatrix} 1 - j\frac{m_2M_1}{m_1\sigma^2}\tan(kl_2) & \left(1 - \frac{1}{\sigma^2}\right)\frac{M_1}{m_1} \\ jm_2\tan(kl_2) & 1 + \alpha_2 \end{bmatrix} \tag{12}$$

where $\alpha_2 = j\frac{m_2M_1}{m_1}\tan(kl_2)$.

Similarly, the second chamber can be represented by the transfer matrix method as a combination of the insertion tube sudden expansion unit, the straight tube unit and the insertion tube sudden contraction unit. The third chamber can also be expressed by the transfer matrix method as a combination of the insertion tube sudden expansion unit, the straight tube unit and the insertion tube sudden contraction unit. The total transfer matrix of the designed muffler can be expressed as:

$$T = T_1 \cdot T_2 \cdot T_3 \cdot T_4 \cdot T_5 \cdot T_6 \cdot T_7 \cdot T_8 \cdot T_9 = \begin{bmatrix} A & B \\ C & D \end{bmatrix} \tag{13}$$

Then, the transmission loss equation for the three-cavity interpolated perforated muffler is:

$$TL(f) = 10\lg\left(\frac{|A + m_3B + C + m_3D|^2}{4m_3}\right) \tag{14}$$

where $k$ is the number of waves, $k = \omega/c$; $\omega$ is the angular velocity; $f$ is the frequency; $c$ is the speed of sound; $m$ is the ratio coefficient of the area of the suction pipe to the main pipe; $m_1$ is the area ratio coefficient of exhaust pipe to main pipe; $m_2$ is the area ratio coefficient of cavity to main pipe; $m_3$ is the ratio of the cross-sectional areas of the muffler suction and exhaust port ends; $\sigma$ is the generalized expansion ratio; $M$ is the Mach number of airflow in a straight pipe; $M_1$ is the Mach number of airflow in a thin pipe.

The transmission loss equation for the full frequency range (20–5000 Hz) of the three-cavity perforated muffler is selected. To achieve the maximum transmission loss, the optimization objective function is defined as:

$$\text{Max } f(x) = -\frac{1}{f_2 - f_1} \int_{f_1}^{f_2} TL(f) \cdot df \tag{15}$$

where $f(x)$ is the optimization objective;

$x$ is the optimization variables;
$f_1$ is the lower limit of the objective function frequency (20 Hz);
$f_2$ is the objective function frequency upper limit (5000 Hz);
$TL(f)$ is the transfer loss theoretical equation.

(1) Optimization variables and constraints

According to muffler noise reduction theory, increasing the muffler cavity diameter and the total length of the cavity generally improves the noise reduction effect. Therefore, within the constraints of the available installation space, the cavity diameter and total length should be maximized. At the same time, due to installation constraints, the diameters of the suction and exhaust pipes, as well as the internal perforated pipes, remain unchanged. Therefore, the design variables are the lengths of the following components: the muffler suction insertion pipe, the first cavity exhaust pipe, the second cavity suction pipe, the second cavity exhaust pipe, the third cavity suction pipe, the muffler exhaust insertion pipe, the first cavity length and the second cavity length. These variables are denoted as $x_1$, $x_2$, $x_3$, $x_4$, $x_5$, $x_6$, $x_7$ and $x_8$, respectively.

$$x = [x_1, x_2, x_3, x_4, x_5, x_6, x_7, x_8]^{\text{T}} \tag{16}$$

(2) Constraint conditions

According to the relationship between the structural dimensions of the muffler, the constraint conditions can be obtained as:

$$\text{s.t.} \begin{cases} g_1(x) = l_7 + l_8 < 190 \text{ mm} \\ g_2(x) = l_7 + l_8 - l_9 < -10 \\ g_3(x) = l_1 + l_2 < l_7 \\ g_4(x) = l_3 + l_4 < l_8 \\ g_5(x) = l_5 + l_6 < l_9 - 10 - l_7 - l_8 \\ g_6(x) = l_1 + l_2 + l_3 + l_4 + l_5 + l_6 + 10 < 200 \text{ mm} \\ u_i < x_i < v_i \end{cases} \tag{17}$$

where $u_i$ is the lower limit of the optimization variable; $v_i$ is the upper limit of the optimization variable.

(3) Optimization results and analysis

Through the structural optimization of the perforated muffler, the values of each structural parameter were determined to maximize the muffler's transmission loss. The structural parameters of the muffler before and after optimization are presented in Table 2.

**Table 2.** Structural parameters of three-cavity interpolated perforated muffler before and after optimization.

| Variable | Lower Limit of Variable/mm | Upper Limit of Variable/mm | Initial Value/mm | Optimization Parameter Value/mm |
|---|---|---|---|---|
| the muffler suction insertion pipe length | 0 | 190 | 30 | 21.2 |
| the first cavity exhaust pipe length | 0 | 190 | 17.5 | 13.3 |
| the second cavity suction pipe length | 0 | 190 | 17.5 | 23.6 |
| the second cavity exhaust pipe length | 0 | 190 | 17.5 | 19.8 |
| the third cavity suction pipe length | 0 | 190 | 17.5 | 16.7 |
| the muffler exhaust insertion pipe length | 0 | 190 | 30 | 27.9 |
| the muffler first cavity length | 0 | 190 | 63.3 | 50.8 |
| the muffler second cavity length | 5 | 195 | 63.3 | 68.9 |

Using COMSOL software, the transmission loss of the structurally optimized muffler is calculated and compared with the transmission loss of the muffler before optimization, as shown in Figure 20.

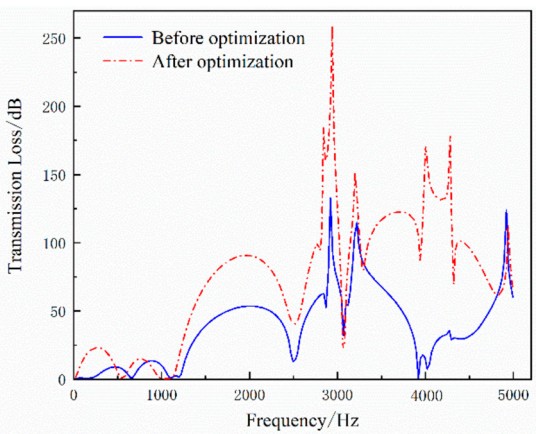

**Figure 20.** Comparison of transmission loss before and after optimization of three-cavity perforated muffler.

As shown in Figure 17, the transmission loss of the three-cavity perforated muffler is significantly improved after optimization. Notably, between 1200 Hz and 5000 Hz, the transmission loss of the optimized muffler is markedly higher compared to the pre-optimization values. Additionally, the cut-off frequency phenomenon in the mid- and high-frequency bands is notably reduced, and the issue of zero transmission loss at 3920 Hz is resolved. The transmission loss at frequencies of 2500 Hz and 4000 Hz has also been enhanced, leading to improved overall noise reduction performance of the muffler.

Although the optimized muffler exhibits improved noise reduction in the mid- and high-frequency bands, it still has low transmission loss in the low-frequency band, resulting in poor and unstable noise reduction performance. To enhance the stability of noise reduction, sound-absorbing material is added to the inner-wall surfaces of the muffler. Glass fiber, with a thickness of 10 mm, is chosen as the sound-absorbing material. Figure 21 illustrates the transmission loss of the muffler before and after this structural optimization.

As shown in Figure 18, the addition of sound-absorbing materials significantly improves the muffler's transmission loss in the low-frequency band. The fluctuation and cut-off frequency phenomenon are eliminated, and the transmission loss value steadily increases with frequency, resulting in stable noise reduction performance. In the mid- and high-frequency bands, the transfer loss curve becomes less variable, the noise reduction increases and overall noise reduction performance is more stable.

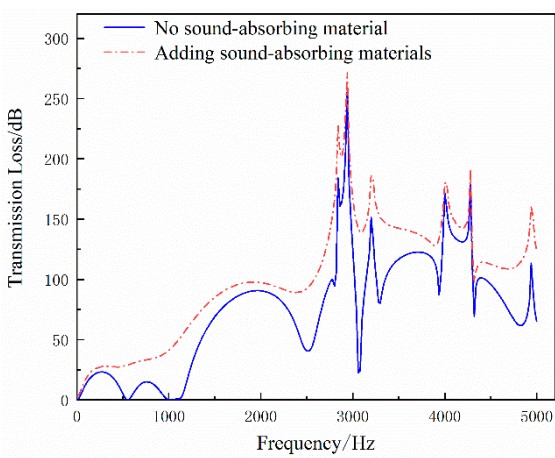

**Figure 21.** Comparison of silencer transmission loss.

### 5.5. Study of Noise Reduction at Air Compressor Exhaust Ports

The designed Helmholtz resonator and the three-cavity perforated muffler were sequentially installed at the exhaust port of the twin-screw air compressor, as shown in Figure 22. With the inlet pressure set to 0.1 MPa, the exhaust pressure to 0.8 MPa, the inlet temperature at 20 °C, the exhaust temperature at 80 °C and the speed of sound at 340 m/s, and assuming a hard acoustic field wall for the shell, noise reduction analysis was performed using multi-physical field coupling calculations. The noise spectra following the installation of the exhaust muffler were then obtained. Since the sound wave propagates as a plane wave in a circular pipe with a constant cross-sectional shape, the sound energy remains unchanged. Therefore, the pipe does not affect the noise at the air compressor exhaust port and will not be considered in this analysis.

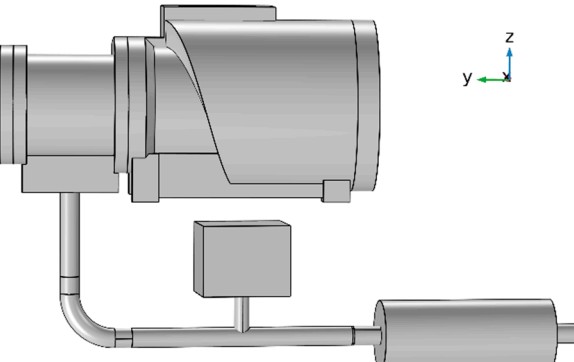

**Figure 22.** Air compressor and exhaust muffler assembly diagram.

Figure 23 illustrates a comparison of noise levels at the exhaust port of the air compressor before and after the installation of the muffler. After installing the exhaust muffler, significant noise reduction is observed. Without the muffler, the noise at the exhaust port ranges from 100 to 114 dB, with a peak of 115 dB. In contrast, with the muffler installed, the noise levels are reduced to a range of 37 to 68 dB, with a maximum of 76.1 dB. This indicates that the muffler significantly reduces noise and maintains stable performance in noise reduction.

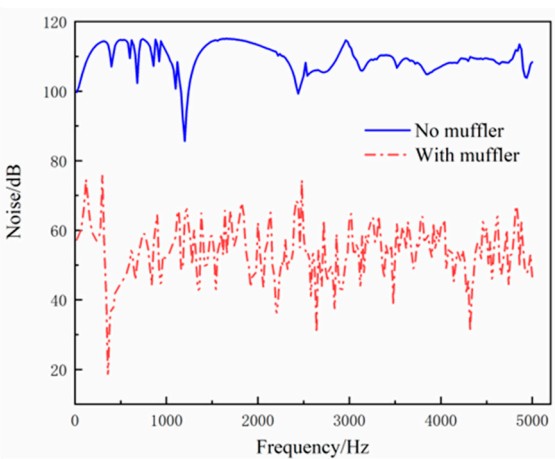

**Figure 23.** Comparison of noise at the exhaust port of air compressor before and after installation of muffler.

### *5.6. The Effect of Mufflers on the Performance of Air Compressors*

To evaluate the effect of the muffler on the exhaust pressure of the twin-screw compressor, measurements were taken after sequentially installing the Helmholtz resonator and the three-cavity interpolated perforated muffler at the exhaust port. The set parameters were a suction pressure of 0.1 MPa, a suction temperature of 20 °C, an exhaust temperature of 80 °C, a sound speed of 340 m/s and a hard acoustic field wall for the casing. The exhaust pressure was measured under a working condition of 0.8 MPa, resulting in a value of 798,175 Pa. The exhaust pressure error was only 0.228%, indicating that while the three-cavity interpolated perforated muffler does cause a slight pressure loss, it is minimal and within acceptable limits for the compressor's operation. Additionally, the muffler demonstrates effective noise reduction, making it suitable for use as a noise reduction solution for the compressor's exhaust.

### 6. Conclusions

In this study, the internal flow field model of the twin-screw air compressor was analyzed to determine the intake and exhaust noise levels. To address the issue of significant noise at the exhaust port, an exhaust muffler was designed, and a noise reduction analysis was conducted. The results showed a marked improvement in noise reduction, successfully achieving the desired outcomes.

(1)  By calculating the internal flow field of the twin-screw compressor, it was found that the noise at the air compressor inlet primarily ranges between 84 and 100 dB, while the noise at the exhaust port is predominantly between 100 and 114 dB. Notably, the noise at the exhaust port is significantly higher than that at the air inlet.

(2)  To address the issue of excessive noise at the exhaust port of the air compressor, a Helmholtz resonator was designed to function as a flow stabilizer. An acoustic fluid interaction analysis revealed that the resonator effectively stabilized gas flow and reduced noise in the low-frequency band. Additionally, a three-cavity interpolated perforated muffler was designed for full-frequency performance analysis, and its structure was optimized, resulting in enhanced noise reduction.

(3)  The designed Helmholtz resonator and three-cavity interpolated perforated muffler were installed sequentially at the exhaust port, followed by a noise reduction analysis. The results show that the exhaust port noise decreased significantly from 100–114 dB to 37–68 dB, demonstrating the effectiveness of the designed exhaust muffler in reducing noise. The findings of this study provide valuable insights for noise reduction in equipment with aerodynamic noise issues.

**Author Contributions:** Conceptualization, Y.H.; data curation, X.H. and L.C.; formal analysis, Y.H., X.H. and J.W.; methodology, Y.H., X.H. and J.W.; project administration, Y.H.; software, X.H. and Y.Z.; writing—original draft, Y.H. and X.H.; writing—review and editing, Y.H., X.H. and Z.R. All authors have read and agreed to the published version of the manuscript.

**Funding:** This research was funded by Shaanxi Science Technology Department Key Project of China, grant number [2017ZDXM-GY-138].

**Data Availability Statement:** All data generated and analyzed during this study are included in this published article.

**Conflicts of Interest:** The authors declare no conflicts of interest.

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
