# Peer review of "Flow Field Noise Analysis and Noise Reduction Research of Twin-Screw Air Compressor Based on Multi-Field Coupling Technology"

_machines, doi:10.3390/machines12080577_

Round 1

Reviewer 1 Report

Comments and Suggestions for Authors

The paper shows the application of the noise reducer of the double compressor in a very original way. The authors presented the obtained results analytically, graphically and visually. The paper can be accepted, but it is necessary to carry out certain refinements in order to improve the quality of the work and its application for subsequent tests.

It is necessary to consult and study additional literature and review and cite papers from this and similar fields, which are more recent, such as:

-        DOI 10.17559/TV-20220414105757.

-        DOI 10.1038/s41598-023-48362-4.

-        DOI 10.1016/j.apacoust.2020.107383.

-        Abdulrahem, A. A. (2023). ANALYSIS OF THE JET ENGINE NOISE EMISSIONS AT THE INLET AIR TO THE COMPRESSOR AND NOZZLE EXHAUST OUTLET. Journal of Engineering Research18(35), 11-11.

-        DOI 10.17559/TV-20220414105757.

Expand the introduction or other chapters with papers in this area.

As it is an exceptionally high-quality paper, it is necessary to specify at the end of the Introduction of the paper what is the main contribution of the paper and how does this papaer differ from similar papers in this field? What are the main reasons, i.e. why do the authors think that this paper should be published?

Increase the number of references in the paper based on the given remarks.        

Increase the number of references in the paper based on the given remarks, especially bearing in mind that considering the scope of the paper and the importance of the paper, a very small number of papers are cited in it.        

In the text explaining individual equations, it is necessary to indicate whether the mentioned equations were taken from some of the earlier papers and other literature (if they are, it is necessary to cite the literature), or whether the authors came up with the mentioned equations themselves.  

In several places Freqency is written instead of Frequency.

It is necessary to indicate whether the reproducibility of the results was ensured in the experiment and simulation, that is, how many times the experiment or simulation was repeated in order to obtain the mean values ​​of the observed quantities.

In the concluding remarks, it is necessary to clearly state whether the obtained results confirmed the expectations and assumptions, that is, whether there were deviations or completely opposite results from the expected ones.

It is necessary to state in which mechanical constructions it is possible to use the mentioned compressor solutions, as well as further possible directions of development and further improvements, bearing in mind the results and achievements presented in this paper. Look at the possibilities to comment on the reduction or increase in the costs of exploitation and maintenance of assemblies in which applied compressor solutions are used.   

Author Response

Manuscript Title: Flow field noise analysis and noise reduction research of twin-screw air compressor based on multi-field coupling technology

Paper No: machines-3064107                                                

Authors:Yayin He*, Xuyang He, Lijun Chen, Junli Wang, Yongqiang Zhao and Zhigui Ren     

Responses to the Reviewers

We would like to thank the editor and the reviewers for the constructive reviews of our above-mentioned paper (No:machines-3064107), and the valuable suggestions for further revisions. We have carefully modified this paper based on the comments. The point-by-point response to the questions and the corresponding revisions are summarized below. We hope the revised manuscript is acceptable for publication.

Comments of Reviewer

The paper shows the application of the noise reducer of the double compressor in a very original way. The authors presented the obtained results analytically, graphically and visually. The paper can be accepted, but it is necessary to carry out certain refinements in order to improve the quality of the work and its application for subsequent tests.

Comment 1

It is necessary to consult and study additional literature and review and cite papers from this and similar fields, which are more recent, such as:

-        DOI 10.17559/TV-20220414105757. 

-        DOI 10.1038/s41598-023-48362-4.

-        DOI 10.1016/j.apacoust.2020.107383.

-        Abdulrahem, A. A. (2023). ANALYSIS OF THE JET ENGINE NOISE EMISSIONS AT THE INLET AIR TO THE COMPRESSOR AND NOZZLE EXHAUST OUTLET. Journal of Engineering Research, 18(35), 11-11.

-        DOI 10.17559/TV-20220414105757.

Expand the introduction or other chapters with papers in this area.

Response to the comment 1

The authors have reviewed more literature, including several papers presented by the reviewers, which have been used as references and added to the introduction.

Comment 2

As it is an exceptionally high-quality paper, it is necessary to specify at the end of the Introduction of the paper what is the main contribution of the paper and how does this papaer differ from similar papers in this field? What are the main reasons, i.e. why do the authors think that this paper should be published?

Response to the comment 2

The article has added the shortcomings of the existing research and the main innovations of the paper at the end of the introduction.

Comment 3

Increase the number of references in the paper based on the given remarks.    

Response to the comment 3

The number of references has been increased.

Comment 4

Increase the number of references in the paper based on the given remarks, especially bearing in mind that considering the scope of the paper and the importance of the paper, a very small number of papers are cited in it.       

Response to the comment 4

The number of references has been increased.

Comment 5

In the text explaining individual equations, it is necessary to indicate whether the mentioned equations were taken from some of the earlier papers and other literature (if they are, it is necessary to cite the literature), or whether the authors came up with the mentioned equations themselves.

Response to the comment 5

The equations applied in the theoretical part of the article are cited from other literature, and have been cited in the corresponding part of the article.

Comment 6

In several places Freqency is written instead of Frequency.

Response to the comment 6

Modifications have been made.

Comment 7

It is necessary to indicate whether the reproducibility of the results was ensured in the experiment and simulation, that is, how many times the experiment or simulation was repeated in order to obtain the mean values ​​of the observed quantities.

Response to the comment 7

In order to ensure the repeatability of the results, the simulation of air compressor flow field noise, muffler performance analysis, performance optimization and noise reduction analysis was repeated three times, and the experiments were measured five times at the same time, and the average value of observations was obtained. These have been supplemented in the article.

Comment 8

In the concluding remarks, it is necessary to clearly state whether the obtained results confirmed the expectations and assumptions, that is, whether there were deviations or completely opposite results from the expected ones.

Response to the comment 8

The results of this study have been met and are supplemented in the text.

Comment 9

It is necessary to state in which mechanical constructions it is possible to use the mentioned compressor solutions, as well as further possible directions of development and further improvements, bearing in mind the results and achievements presented in this paper. Look at the possibilities to comment on the reduction or increase in the costs of exploitation and maintenance of assemblies in which applied compressor solutions are used.  

Response to the comment 9

The research results of this paper can be applied to the aerodynamic noise reduction of screw compressors, and at the same time, it has a reference role for the noise reduction of equipment with aerodynamic noise, such as automobile and motorcycle exhaust systems. For mufflers used in different occasions, mufflers with different structures can be used. It has been added in the conclusion of the article.

Reviewer 2 Report

Comments and Suggestions for Authors

The compressor noise research and reduction has become a measure of the new requirements of compressor technology. The paper analyzed the characteristics of a air compressor intake noise and exhaust noise, and also designed the exhaust muffler, it has good engineering significance. But the paper has the following problems:

1. The introduction of research methods in the paper is not detailed enough,  for example, the multi-physics field coupling software is not introduced. 

2. The calculation accuracy of multi-physics field coupling software is not been verified.

3. In order to verify the correctness of the simulation calculation of the noise, a twin-screw air compressor with the same technical parameters and working conditions as the model constructed in  the paper is used as the test object. Form Figure 8, we can see the electric motor is very close to the compressor, how to eliminate the influence of electric motor noise on measurement results??

4. Due to the air compressor, the air flow pulsation at the sunction port and exhaust port is very strong, but it may not form effective noise radiation, how does the author consider this issue??

5.  It can be seen that the noise generated at the exhaust port of the compressor is significantly larger than the noise of the suction port. Why not measure the noise at the exhaust?

6. By comparing with the air compressor without muffler, it is found that the noise atthe exhaust port is reduced from 100-114 dB to 37-68 dB. It is very good, but how to verify the credibility of this result?

7. Some formula numbers in the paper are incorrect. In Chapter 2.2 of the paper, "the equations for plane and three-dimensional waves in a moving medium are shown in Eqs. 6 and 7" should be "the equations for plane and three-dimensional waves in a moving medium are shown in Eqs. 7 and 8 " .

Author Response

Manuscript Title: Flow field noise analysis and noise reduction research of twin-screw air compressor based on multi-field coupling technology

Paper No: machines-3064107                                                

Authors:Yayin He*, Xuyang He, Lijun Chen, Junli Wang, Yongqiang Zhao and Zhigui Ren     

Responses to the Reviewers

We would like to thank the editor and the reviewers for the constructive reviews of our above-mentioned paper (No:machines-3064107), and the valuable suggestions for further revisions. We have carefully modified this paper based on the comments. The point-by-point response to the questions and the corresponding revisions are summarized below. We hope the revised manuscript is acceptable for publication.

Comments of Reviewer

The compressor noise research and reduction has become a measure of the new requirements of compressor technology. The paper analyzed the characteristics of a air compressor intake noise and exhaust noise, and also designed the exhaust muffler, it has good engineering significance. But the paper has the following problems:

Comment 1

The introduction of research methods in the paper is not detailed enough,  for example, the multi-physics field coupling software is not introduced. 

Response to the comment 1

Complements have been made to multiphysics coupling software and research methods.

Comment 2

 The calculation accuracy of multi-physics field coupling software is not been verified.

Response to the comment 2

In this paper, the multiphysics coupling software COMSOL is used to calculate the flow field noise and reduce the noise of the air compressor, and the multiphysics coupling modules of "Pressure Acoustics-Frequency Domain" and "Acoustic Flow Coupling" are used for calculation. For the calculation steps, please refer to the solution method and procedure of the case Modeling Vibration and Noise in a Gearbox:BearingVersion"(https://cn.comsol.com/model/modeling-vibration-and-noise-in-a-gearbox-47841) on the COMSOL website. The accuracy of the calculations and results in this case has been confirmed, so the accuracy of the multiphysics calculations has been confirmed.

Comment 3

In order to verify the correctness of the simulation calculation of the noise, a twin-screw air compressor with the same technical parameters and working conditions as the model constructed in  the paper is used as the test object. Form Figure 8, we can see the electric motor is very close to the compressor, how to eliminate the influence of electric motor noise on measurement results?

Response to the comment 3

TIn the twin-screw air compressor unit, the noise source is mainly mechanical noise and suction and exhaust noise, while the influence of motor noise is relatively small, and the measured position is 0.2 meters above the suction port, which is far away from the motor, and the influence of motor noise is relatively small, so it is not considered in this paper.

Comment 4

Due to the air compressor, the air flow pulsation at the sunction port and exhaust port is very strong, but it may not form effective noise radiation, how does the author consider this issue??

Response to the comment 4

For the phenomenon that the air flow pulsation is very strong but there is no noise radiation at the suction port, it is possible that a silencing device, pipe structure, shell sound insulation and other devices are used at the suction port (for details, please refer to DOI: 10.16051/j.cnki.ysjjs.2023.03.005), and the suction port of this experimental equipment does not adopt the above-mentioned additional components, so it can form effective noise.

Comment 5

 It can be seen that the noise generated at the exhaust port of the compressor is significantly larger than the noise of the suction port. Why not measure the noise at the exhaust?

Response to the comment 5

Because the exhaust port of the field experimental equipment is installed with exhaust pipes and oil-gas separation devices, etc., it is not convenient for noise measurement; In addition, the noise calculation at the suction port and the exhaust port is performed in the same way, so the noise at the suction port is measured to verify the accuracy of the simulation calculation.

Comment 6

By comparing with the air compressor without muffler, it is found that the noise atthe exhaust port is reduced from 100-114 dB to 37-68 dB. It is very good, but how to verify the credibility of this result?

Response to the comment 6

The simulation of the muffler transmission loss in this article is based on the calculation steps of the Absorptive Muffler on the COMSOL website (https://cn.comsol.com/model/absorptive-muffler-1367). Adding a muffler to the suction or exhaust port of the air compressor is a very effective way to reduce noise,Such as the literature (DOI:10.27234/d.cnki.gnhuu.2021.000993. Simulation and noice reduction of electromagnetic and pneumatic noice in electric scroll compresso), mainly by adding an expanded muffler to the exhaust port to reduce the noise of the air compressor, and the correctness of the method is verified by experiments. In this paper, a three-chamber perforated muffler is added to the exhaust port for noise reduction, and the simulation calculation is also based on the calculation steps of the Absorptive Muffler on the COMSOL official website, but due to the limitations of the factory experimental conditions, the muffler is not installed for experimental verification.

Comment 7

Some formula numbers in the paper are incorrect. In Chapter 2.2 of the paper, "the equations for plane and three-dimensional waves in a moving medium are shown in Eqs. 6 and 7" should be "the equations for plane and three-dimensional waves in a moving medium are shown in Eqs. 7 and 8 " .

Response to the comment 7

Modifications have been made.

Reviewer 3 Report

Comments and Suggestions for Authors

Good

Author Response

The reviewer did not ask questions about the article.

Round 2

Reviewer 1 Report

Comments and Suggestions for Authors

Thank you for accepting the suggestions.

Author Response

感谢您对本文的感谢。

Reviewer 2 Report

Comments and Suggestions for Authors

The author added the introduction of research methods, which further improved the quality of the paper. However, it is suggested to make it clear that COMSOL software is used in this paper in terms of the introduction of research methods. 

In the introduction of research methods in 3.1 and 3.2 of this paper, many descriptions are rough, and it is recommended to directly clarify that COMSOL software is used in this paper  and point out the software version.In this way, neither the calculation method nor the calculation accuracy need to be described in detail.

Author Response

Thanks to your suggestion to the article, the author has added the software COMSOL and its version in Section 3.2, i.e. COMSOL5.6.